# From Housewives to Employees, the Mental Benefits of Employment across Women with Different Gender Role Attitudes and Parenthood Status

**DOI:** 10.3390/ijerph20054364

**Published:** 2023-02-28

**Authors:** Zhuofei Lu, Shuo Yan, Jeff Jones, Yucheng He, Qigen She

**Affiliations:** 1Department of Social Statistics, University of Manchester, HBS Building, Oxford Road, Manchester M13 9PL, UK; 2Warwick Manufacturing Group, The University of Warwick, Coventry CV4 7AL, UK; 3Independent Researcher, Chongqing 400000, China; 4Faculty of Science and Technology, Beijing Normal University-Hong Kong Baptist University United International College (UIC), Zhuhai 519000, China

**Keywords:** employment, gender role attitudes, housewives, mental health, parenthood

## Abstract

Previous studies suggest that paid employment can improve workers’ mental health status by offering a series of manifest and latent benefits (i.e., income, self-achievement and social engagement), which motivates policymakers’ ongoing promotion of labour force participation as an approach to protect women’s mental health status. This study extends the literature by investigating the mental health consequences of housewives’ transition into paid employment across different gender role attitude groups. In addition, the study also tests the potential moderating role of the presence of children in relationships. This study yields two major findings by using nationally representative data (N = 1222) from the United Kingdom Longitudinal Household Study (2010–2014) and OLS regressions. First, from the first wave to the next, housewives who transitioned into paid employment reported better mental health status than those who remained housewives. Second, the presence of children can moderate such associations, but only among housewives with more traditional gender role attitudes. Specifically, among the traditional group, the mental benefits of transition into paid employment are more pronounced among those without children. Therefore, policymakers should develop more innovative approaches to promote housewives’ mental health by considering a more gender-role-attitudes-sensitive design of future labour market policies.

## 1. Introduction

During the last decades, women have increasingly participated in the labour market, European countries have actively promoted gender equality in education and the labour market to protect women’s rights [1]. In addition, there has been a continuous gender convergence in the division of housework, with women reducing their unpaid work and transitioning into paid employment [2]. Although there still remain gender pay gaps and ‘glass ceilings’ in the UK labour market [3], many scholars believe that these profound changes in women’s labour force participation have improved women’s well-being [4,5]. However, recent studies have challenged that women will be more satisfied with being housewives or working part-time than being in full-time employment, especially when they have childcare responsibilities. This is because women are under pressure to have a career [6]. Indeed, employed women’s mental well-being has remained at a low level during the last decades in the UK [7], which motivates an exploration of the nuanced understanding of the mental health differences between housewives and employed women, and an investigation into the roles of more socio-demographic factors.

There are conflicting theoretical predictions and empirical findings about how transitioning into paid employment affect housewives’ mental health. On the one hand, employment sociologists argue that paid employment is beneficial to women’s subjective well-being by ensuring both manifest benefits (i.e., income) and latent benefits (i.e., social engagement and self-identity development) [8,9]. In addition, some scholars claim that career women’s achievement in their paid work can buffer the work–family conflicts they suffer, thereby increasing women’s mental health [10]. On the other hand, family and gender sociologists argue that women are not always satisfied with participating in the labour market. This is because paid employment makes them lose their chance and time with their families, especially for women with more traditional gender role attitudes and with young children [6,11].

The existing studies are limited in several aspects, which may contribute to inconsistent predictions and findings. First, while studies indicate that gender role attitudes are crucial in the debates on the associations between employment and women’s mental health [12,13], the investigation of gender role attitudes is absent in most studies on the debates around the mental health consequences of women’s paid employment and housewives’ status. Specifically, gender role attitudes refers to a person’s attitudes to the division of labour in the household [12,14]. For instance, women with more traditional gender role attitudes tend to accept the norms that expect women to take on more household responsibilities and that men should be the breadwinners [15]. By contrast, women with egalitarian gender role attitudes tend to reject the traditional gendered division of labour. Thus, failing to consider the role of women’s gender role attitudes prevents us from gaining a nuanced understanding of the within-gender differences. Second, while many studies show that motherhood and the number of children can reshape women’s gender role attitudes during and after giving birth [16,17,18,19], current studies on the associations between women’s employment and mental health ignore the role of the presence of children. The potential interactions between employment, gender role attitudes and the presence of children are under investigated. Therefore, it is necessary to reinvestigate the impacts of housewives’ transition to paid employment on their mental health by integrating the literature on employment, gender role attitudes and the presence of children.

### 1.1. Theoretical Bases

This section aims to demonstrate how the transition from housewives to paid employment shapes women’s mental health. There is a research tradition highlighting the benefits of employment to mental health [20,21]. In the first place, Jahoda’s Latent Deprivation Theory [8,9] and Fryer’s Agency Restriction thesis [9] have strongly emphasised the benefits of employment and discussed how paid employment shapes workers’ mental health [22]. Specifically, Jahoda (1982) argued that paid employment could supply not only manifest benefits (i.e., financial resources) but also latent benefits (i.e., self-achievement, social engagement and identity development, etc.), thereby protecting workers’ mental health. During the last decade, a strand of empirical studies have tested the theoretical predictions of Jahoda’s Latent Deprivation Theory, for instance, many studies found that unemployed women and women doing unpaid work report worse mental health than those who were in paid work, regardless of the use of flexible working arrangements [5,23,24]. In the second place, the ‘role expansion’ hypothesis indicates that an individual’s experience in their work (i.e., as an employee) can improve their performance and satisfaction in the other roles (i.e., as a mother) [10]. For instance, a stream of studies on ‘positive spill over’ from work to family suggests the ‘role expansion’ hypothesis, indicating that achievement in the work domain can buffer the stress and conflicts in households [1,25,26]. Thus, housewives might have worse mental health status due to the deprivation of the mental benefits brought on by paid employment.

#### 1.1.1. The Within-Gender Differences: Gender Role Attitudes

Another stream of studies indicates that women’s employment will erode their mental health, especially when their paid work is out of step with their gender role attitudes. The social comparison theory [27] and the doing gender theory [28] indicate that women with more traditional gender role attitudes are more willing to carry out childcare and domestic work, since this is a way to express their femininity and fulfil cultural expectations., Traditional gender role attitudes can be shown by women by their willingness to accept the cultural norms that expect them to take on more household responsibilities. Thus, women with more traditional gender role attitudes tend to suffer more inter-role conflicts when suffering high work demands since the ideal worker norms are always in direct conflict with their family commitments. For example, studies have found that traditional gender role attitudes might reduce women’s employment willingness and probability, especially for mothers with young children [11,29]. In addition, previous studies have also found that women with more traditional gender role attitudes are more vulnerable to workload and work inflexibility [13,14]. Thus, scholars believe that gender role attitudes are one of the most important factors in predicting career women’s mental health, and the impacts of housewives’ transition into paid employment might be different between different gender role groups [30]. As for housewives with more traditional gender role attitudes, paid work might not benefit their mental health.

#### 1.1.2. The Presence of Children

Furthermore, the presence of children in the household can be a potential moderator in the associations between housewives’ employment transition and mental health. This is because the division of household labour is constrained by women’s intrinsically time-consuming and energetically demanding responsibilities (i.e., childbearing and nursing infants), which prevents women from taking on other tasks [19,31]. For instance, studies indicate that the amount of time that women spend on domestic work and childcare will significantly increase after transitioning into parenthood, while this pattern is not always true for men [32,33,34]. In addition, women with insufficient time for childcare report more mental issues [35]. Accordingly, employment might be harmful to their mental health by reducing their time with their families. It is also worth noting that there might be an intersection of women’s gender roles and the presence of children. This is because housewives with more traditional gender role attitudes are more likely to have increased domestic demands when having children [36,37], thereby suffering more work–family conflicts when doing paid work [38,39]. Taken together, it can be assumed that the transition to paid employment would generally benefit women’s mental health. However, such impacts will be differentiated by gender roles, the presence of children or the combination of the two [38,39,40,41]. Specifically, housewives with more traditional gender role attitudes will benefit less from the transition to employment than those with more egalitarian attitudes. In addition, housewives with children will benefit less than those without children.

### 1.2. Current Study

To address these research gaps, the literature on employment, gender role attitudes and the presence of children was integrated. Using nationally representative household survey data in the UK (2010–2014), an investigation into how transitioning into paid employment shapes housewives’ mental health across women with different gender role attitudes and parenthood status was undertaken. Specifically, the investigation targets the potential within-gender differences (gender role attitudes) by conducting analyses with separate samples: (i) traditional women and (ii) egalitarian women. In addition, the potential moderating role of the presence of children in the associations was tested. Overall, this study finds that women’s transition from housewives to paid employment, gender role attitudes and the presence of children interact to shape mental health. Specifically, housewives who transitioned into paid employment tended to have better mental health status than those who remained to be housewives. In addition, the presence of children can moderate such associations but only among the group with traditional gender role attitudes. Among housewives with more traditional gender role attitudes, the positive association between transition into employment and mental health is more pronounced among those without children. Therefore, this study suggests that policymakers should promote more innovative approaches to protect specific groups (i.e., traditional housewives with the presence of children) since they are more vulnerable to the potential adverse mental effects of paid employment and work–family conflicts.

## 2. Methodology

### 2.1. Data and Sample

The United Kingdom Household Longitudinal Study (UKHLS) is the largest household survey in the UK, which uses a stratified and clustered sampling design to collect a sample of around 40,000 households in the first wave (2009) [42]. This study uses the second (2010–2012) and the fourth (2012–2014) waves of the (UKHLS), as these two waves contain consistent measures of respondents’ gender role attitudes and job statuses [43]. The most recent waves encompassing the COVID-19 period were not used. This is because only the second, fourth, and tenth waves of the UKHLS contain information about the respondents’ gender role attitudes, and the time interval between the second (2010–2012) and the fourth (2012–2014) waves is within a short period [42,44]. The sample of this study is restricted to women who reported being housewives in the second wave, and then transitioned to paid employment (including self-employment) or remained to be housewives in the fourth wave. The final analytic sample includes 1222 respondents. See Figure 1 for details about the analytic sample construction process.

### 2.2. Measures

Dependent variable: The dependent variable of this study is the women’s mental health status, which is measured by the 12-item General Health Questionnaire (GHQ-12), a widely used scale for assessing mental health [45]. In the UKHLS, the GHQ-12 responses were transformed into a single continuous scale, ranging from 0 (the least distressed) to 36 (the most distressed) [46]. The data about women’s GHQ are drawn from wave 4. For the sake of clarity, this research follows the practice of previous research [47] and reverses the scale, thus, in this study, a higher score of GHQ indicates better mental health status.

Independent variables: Women’s job status is the main independent variable in this study, which is dichotomised into (i) women who remain to be housewives from wave two to wave four and (ii) women who transition from being housewives in wave two to paid employment, including being self-employed in wave four. Specifically, during the survey, respondents recorded their job status in each wave [45]. Thus, to generate the variable of the women’s job status transitioning in this study, firstly, women who were reported as being housewives (doing family care or housework but no paid work) in wave two were kept as the target observations in this study. Secondly, the job status of these target observations in the fourth wave was tracked, and the cases who remained to be housewives or transitioned to paid employment in wave four were kept. Overall, the data about women’s job status are drawn and generated from both wave two and wave four.

Gender Role Attitudes (GRA): In this study, women’s Gender Role Attitude (GRA) is measured by four Likert-type statements capturing women’s attitudes towards the division of domestic and financial responsibilities [48]. Specifically, during the survey in the UKHLS, respondents were asked to rate their attitude towards four statements on a five-point scale, ranging from 1 “strongly agree” to 5 “strongly disagree” [42]. The four statements are (i) “all in all, family life suffers when the woman has a full-time job”; (ii) “a husband’s job is to earn money, a wife’s job is to look after the home and family”; (iii) “a pre-school child is likely to suffer if his or her mother works”; (iv) “both the husband and wife should contribute to the household income”. Where necessary (that is, the fourth statement), the responses were reversed so that “strongly agree” reflects more traditional gender role attitudes, and the answer of “strongly disagree” reflects more egalitarian gender role attitudes [18,44]. In this study, firstly, the respondents’ ratings on the four statements were summed and then divided by four for data in both wave two and wave four. Then respondents’ average GRA scores in waves two and four were used to generate a new five-point scale variable. Finally, to ease interpretation and statistical modelling, GRA was dichotomised as a binary variable, and 2.5 of 5.0 is the cut-off point of this variable. Hence, the groups are (i) traditional groups (with scores lower than or equal to 2.5 of GRA) and (ii) egalitarian groups (with scores higher than 2.5 of GRA).

Moderator: The presence of children is the moderator in this study, which measures whether the women have dependent children under 16 or not in the household. In the UKHLS, respondents were asked to answer the statement about “the number of own children in the household, including natural children, adopted children and step-children, under the age of 16” [43]. In the UK, 16-year-olds lose a variety of child benefits and begin to be recognised as adults. Hence, a cut-off age of 16 years was chosen in the dataset [49]. In this study, a new variable was generated named the presence of children, (i) “yes” for the presence of children means that the number of children in this household is greater than or equal to one and (ii) “no” for the presence of children means that the number of children in this family was recorded as zero.

Control Variables: In addition to the independent and moderator variables, this study also controls for a number of socio-demographic and job characteristics to exclude potential confounding effects. Current relevant studies have identified a series of potential confounders. For example, marital status and the number of children might be associated with women’s mental health [50,51,52,53]. Therefore, the study controls the number of children and marital status in the analyses. In addition, following previous studies [5,21], gross personal income, the presence of longstanding illness and ethnicity are also selected as potential confounders.

### 2.3. Analytic Strategy

This study applies ordinary least squares regressions (OLS) to investigate the impacts of the transitioning from being a housewife to paid employment on mental health and the moderating role of the presence of children. To investigate the within-gender differences, the analyses was repeated using total samples and separate samples with traditional women and egalitarian women groups. To reflect the intricate UKHLS sampling design and produce representative estimates, all of the presented estimates are weighted properly. By analysing the variance inflation factor, it is noted that all of the study’s models passed the tests of multicollinearity (the VIF scores of all the variables in the models are smaller than 1.5).

## 3. Results

### 3.1. Effects of Transitioning into PAID Employment on Women’s Mental Health Status

Table 1 below shows the weighted sample descriptive statistics in this study, and Table 2 displays the results of a series of multivariate linear regressions, predicting the impacts of transitioning from being a housewife to paid employment on women’s mental health. Specifically, Model 1 in Table 2 shows that compared with women who remained to be housewives, women who transitioned from being housewives to paid employment have significantly better mental health status (coefficient = 1.67, SE = 0.42, *p* < 0.001). Next, the potential within-gender differences was investigated by analysing the impacts of transitioning into paid employment by different gender ideology groups. As shown in Table 2, the results of Model 2 and Model 3 demonstrate that the impacts of the transitioning from being housewives to paid employment on women’s mental health both are significant among the women with traditional gender role attitudes (coefficient = 1.89, SE = 0.87, *p* < 0.05) and the women with egalitarian gender role attitudes (coefficient = 1.68, SE = 0.48, *p* < 0.001). Taken together, the findings of the empirical analyses generally suggest that transitioning into paid employment from being a housewife can significantly benefit the mental health of women.

### 3.2. Potential Moderation Effects of Parenthood (the Presence of Children)

Furthermore, this study investigated the potential moderation effects of the presence of children on the impacts of women’s job status transitioning from being housewives into paid employment on mental health. Firstly, the moderation effects with the total sample was investigated. The results of Model 1 in Table 3 show that there is a significant interaction effect between women’s job status transitions and the presence of children (coefficient = −2.23, SE = 1.09, *p* < 0.05) among the total sample. Secondly, the investigation of the moderation effects by using separate samples (separate by gender ideology groups) was repeated. As shown in Table 3, the results of Model 2 indicate a significant moderation effect of the presence of children in the impacts of women’s job status when transitioning from being housewives into paid employment, among women with more traditional gender role attitudes (coefficient = −4.57, SE = 2.12, *p* < 0.05). By contrast, the results of Model 3 in Table 3 do not suggest such moderation effects of the presence of children (coefficient = −1.85, SE = 1.18, *p* > 0.05).

To better understand the moderating role of the presence of children in the impacts of transitioning into paid employment, the marginal effects were plotted in Figure 2 with the total sample, and Figure 3 with separate samples of the traditional and egalitarian groups. First, as shown in Figure 2, for women’s mental health, both women with and without children can benefit from transitioning from being housewives into paid employment, while the positive impacts of transitioning from being housewives into paid employment are more pronounced among women without children. Second, as shown on the left side of Figure 3 (traditional group), for women’s mental health, women with children can benefit less from transitioning into paid employment than women without children. Third, as shown on the right side of Figure 3 (egalitarian group), similarly to the left side, women with children can benefit less from transitioning into paid employment than women without children. However, the regression results do not suggest such moderation effects of the presence of children in the impact of job status transition among the egalitarian group.

## 4. Discussion

Although plenty of studies have indicated the benefits of women’s labour force participation to their mental health, there is a lack of empirical evidence on identifying the mental consequences of transitioning from being housewives to paid employees among women. Moreover, the roles of gender role attitudes and the presence of children are absent in current literature on the impacts of employment status transition on mental health. This study uses a nationally representative sample of 1222 British women from 2010 to 2014 and ordinary least square regressions to examine how transitioning into paid employment shape housewives’ mental health. In addition, it tests the potential within-gender differences in terms of gender role attitudes in the associations. Moreover, the study investigates the potential moderation effects of the presence of children in such impacts. Overall, this study has yielded the following important findings.

First, this study finds that from the first wave to the next, women whose job statuses have transitioned from housewives to employees tend to have significantly better mental health than those who remain to be housewives. This finding is consistent with the predictions from Jahoda’s Latent Deprivation Theory [8] and Fryer’s Agency Restriction thesis [9], indicating that employment ensures a series of latent benefits (i.e., self-achievement, social engagement and identity development, etc.), thereby improving women’s mental health status. Staying being a housewife may harm women’s mental health due to the deprivation of such benefits and increased domestic work. Moreover, the mental benefits of employment remain significant even among women with more traditional gender role attitudes, which is inconsistent with the predictions from previous studies on gender roles [14] and the doing gender theory [28]. This might be because of the absence of the interaction terms between employment status and the presence of children in the analyses, concealing the nuanced differences within traditional women. Thus, the study further investigates the potential moderation effects of the presence of children with separate samples (traditional group vs. egalitarian group).

Second, this study finds that the presence of children can significantly moderate the impacts of transitioning into employment among housewives with more traditional gender role attitudes but not for their counterparts. Specifically, as for housewives with more traditional gender role attitudes, the benefits of transitioning into employment are more pronounced among those without children. By contrast, housewives with children and more traditional gender role attitudes can hardly benefit from transitioning into employment. Thus, the study’s findings are generally consistent with the theoretical arguments from the doing gender theory [28] and the studies on gender roles and parenthood [5,23]. Housewives with more traditional gender role attitudes and the presence of children can hardly benefit from transitioning into employment due to the persistent institutional gender norms and cultural restrictions. By contrast, as for housewives with more egalitarian gender role attitudes, the moderation effects of the presence of children are insignificant due to their more job-oriented preferences [11,54].

This study has some limitations, which could be potential directions for future studies. Firstly, the study uses the second and fourth waves (2010–2014) of the UKHLS but not the latest waves that covered the COVID-19 period. This is because only the second, fourth and tenth waves of UKHLS collected respondents’ answers to their gender role attitudes. To capture the transition process, two waves of data within a minor time gap were needed, while there is a huge time gap between the fourth and the tenth waves. Future studies can use the upcoming waves of UKHLS to conduct cross-sectional or longitudinal studies to investigate the impacts of transition into employment, considering COVID-19 effects and longitudinal estimations. Secondly, this study focuses on the effects of transitioning into employment on housewives’ individual-level mental health status. However, a growing body of literature highlights the linked lives between different family members within households (e.g., couples and children) [17,55]. Thus, future research should examine the spill over effects of housewives’ transition into employment on mental health across different family members, which could further expand the understanding of the social consequences of housewives’ employment status. Thirdly, since there are still many unobserved confounders, this study is not able to identify the causal relationship between housewives’ employment status and mental health. These limitations should not, however, overshadow this study’s novel contributions to our understanding of the benefits of transitioning into employment on housewives’ mental health across different gender role attitude groups and the role of the presence of children.

## 5. Conclusions

Overall, apart from traditional housewives with the presence of children, transition into employment can significantly improve housewives’ mental health status. Therefore, it is necessary to promote women’s labour participation as a social interference for protecting the mental health of women. However, policymakers should develop innovative approaches to protect specific groups (i.e., traditional housewives with the presence of children) since they are more vulnerable to the potential adverse mental effects of paid employment and work–family conflicts. The nuanced evidence of the study on the role of the presence of children should be taken into account when promoting housewives’ labour force engagement. Public policies should offer more benefits to support housewives with the presence of children, such as by enhancing their access and use of flexible working arrangements and investing more money to build more public day care centres. It is also worth noting that the policy should not focus only on housewives with the presence of children but on the whole family. Overall, governments and organisations should promote more work–family interventions (i.e., longer parental leaves and flexible schedules) to encourage fathers to participate in childcare activities.

## Figures and Tables

**Figure 1 ijerph-20-04364-f001:**
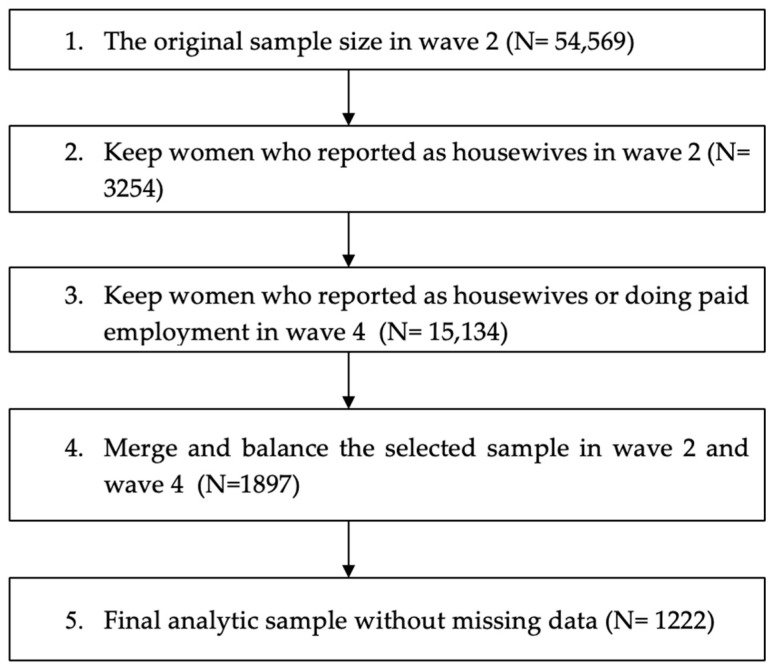
The analytic sample construction process.

**Figure 2 ijerph-20-04364-f002:**
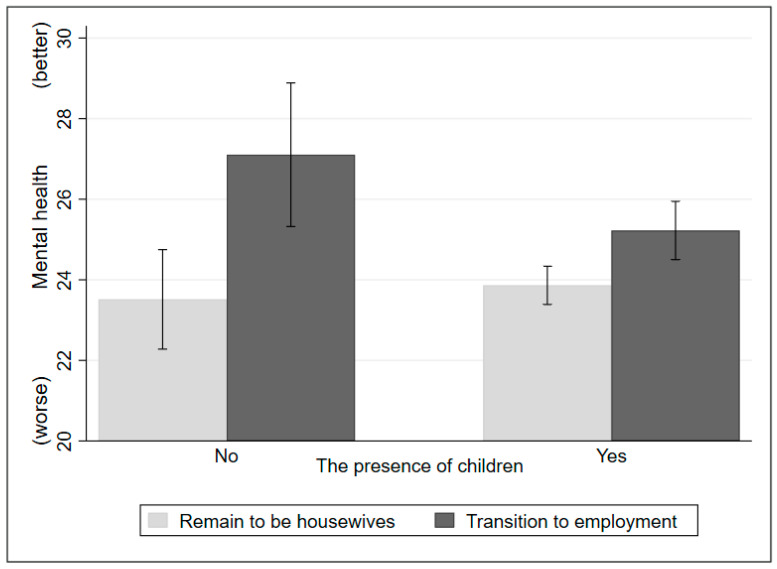
The moderation effects of the presence of children in the impacts of job status transition (total sample).

**Figure 3 ijerph-20-04364-f003:**
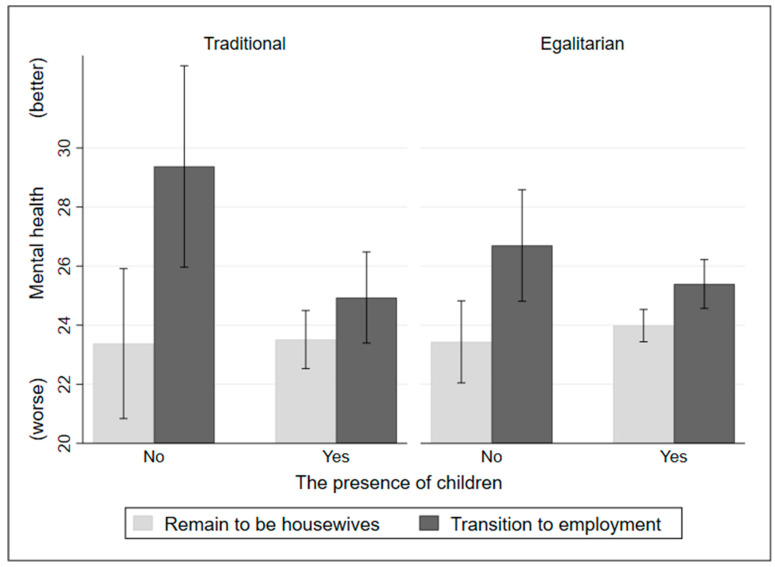
The moderation effects of the presence of children in the impacts of job status transition (separate samples of the traditional and egalitarian groups).

**Table 1 ijerph-20-04364-t001:** Weighted sample descriptive statistics.

Variables	Traditional Women		Egalitarian Women	
	%, Mean	SD	%, Mean	SD
GHQ-12	23.87	6.27	24.32	5.79
Job status, %				
Remain to be housewives	79.19%		73.08%	
Transition into paid employment	20.81%		26.92%	
Age	41.49	8.77	38.93	10.37
Age Groups, %				
18–30 years old	8.85%		25.17%	
31–43 years old	55.19%		41.76%	
43–55 years old	26.81%		25.11%	
56–68 years old	9.15%		7.96%	
Marital status, %				
Never married	15.61%		28.60%	
Married	74.15%		62.39%	
Divorced/separated/widowed	10.24%		9.01%	
Presence of children, %				
No	18.44%		20.12%	
Yes	81.56%		79.88%	
Presence of longstanding illness, %				
Yes	27.27%		28.16%	
No	72.73%		71.84%	
Logged income	5.96	1.66	6.12	1.58
Ethic group				
British	75.22%		83.99%	
Not British	24.78%		16.01%	
Number of respondents (N = 1222)	N = 326 (24.64%)		N = 896 (75.36%)	

Note: % = Proportion, M = Mean, SD = Standard deviation; Weighted analysis.

**Table 2 ijerph-20-04364-t002:** Ordinary least squares (OLS) regressions predicting the impact of transitioning from being a housewife to paid employment on women’s mental health.

GHQ-12	Total Sample	Traditional Women	Egalitarian Women
	Model 1	Model 2	Model 3
Job status (Ref. = Remain to be housewives)			
Transition into paid employment	1.67 ***	1.89 *	1.68 ***
	(0.42)	(0.87)	(0.48)
GRA, (Ref. = Traditional women)			
Egalitarian women	0.58		
	(0.43)		
Presence of children, (Ref. = No)			
Yes	−0.12	−0.45	0.12
	(0.63)	(1.40)	(0.69)
Constant	22.03 ***	18.39 ***	23.04 ***
	(1.36)	(2.87)	(1.39)
Number of respondents (Traditional women)	1222	326	896
R-squared	0.09	0.12	0.10

Note: Robust standard errors in parentheses *** *p* < 0.001, * *p* < 0.05. All models control for gender role attitudes, age group, presence of children, marital status, presence of longstanding illness, logged income and ethnic group. See the Appendix A for all coefficients of the covariates.

**Table 3 ijerph-20-04364-t003:** Ordinary least squares (OLS) regressions predicting the moderation effects of the presence of children in the impact of transitioning from being a housewife to paid employment on women’s mental health.

GHQ-12	Total Sample	Traditional Women	Egalitarian Women
	Model 1	Model 2	Model 3
Job status (Ref. = Remain to be housewives)			
Transition into paid employment	3.59 ***	6.00 **	3.26 **
	(1.02)	(1.96)	(1.10)
Job status × Presence of children, (Ref. = Remain to be housewives × No)			
Transition into paid employment × Yes	−2.23 *	−4.57 *	−1.85
	(1.09)	(2.12)	(1.18)
GRA, (Ref. = Traditional women)			
Egalitarian women	0.58		
	(0.43)		
Presence of children, (Ref. = No)			
Yes	0.35	0.14	0.55
	(0.71)	(1.48)	(0.79)
Constant	21.71 ***	18.05 ***	22.74 ***
	(1.37)	(2.86)	(1.43)
Number of respondents	1222	326	896
R-squared	0.10	0.12	0.10

**Note:** Robust standard errors in parentheses *** *p* < 0.001, ** *p* < 0.01, * *p* < 0.05. All models control for gender role attitudes, age group, presence of children, marital status, presence of longstanding illness, logged income and ethnic group.

## Data Availability

Data are available from an open-access public depository (accessible at https://www.understandingsociety.ac.uk, accessed on 1 December 2022).

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
