# Peer review of "From Housewives to Employees, the Mental Benefits of Employment across Women with Different Gender Role Attitudes and Parenthood Status"

_ijerph, 2023, doi:10.3390/ijerph20054364_

Round 1
Reviewer 1 Report
First, the aim of this study is to investigate the mental consequences of housewives’ transition into paid employment across different gender role attitude groups. Globally, the manuscript is really well written and very pleased to read.
Here are my comments:
Abstract:
It is well structured. Perhaps, authors can detailed the number of participants in the study at this moment.
Introduction:
1. Line 36 : I think this is the “work conditions” of the whole working people (men or women) that have become more employee-friendly. I don’t think there are any labour rights reserved for women.
Literature review:
This part should not exist in this type of epidemiology manuscript. These finding and empirical arguments must be replaced and shortened in the introduction or discussion if necessary.
1. Line 165 : Authors propose 3 general questions :
- I think this is the work objectives which should be detailed by the main objective and the secondary ones.
- There is only 2 research questions detailed.
Methodology:
1. Authors explain us how the sample was made but we don’t know :
a. How people were contacted and how they are responded to?
b. How many total people were excluded from each analysis?
A flow chart could help the readers to understand this point. I hope these points are discussed because 1,222 respondents above 40,000 people this is a very small part of the whole initial sample.
2. This is not usual to introduce a table of results in the methodology section. This table should be replace in the result section. Moreover this table gave use 2 information which should be placed in different results: the weighted sample and some descriptive ones. There is huge differences for two results: age groups and marital status. At this time of the manuscript, we don’t know how the two groups were made: traditional women and egalitarian women.
3. From line 204 to 216: this point is already explained line 185-191. This 2 two must be merged.
4. Line 241: Authors explain us how they dichotomised GRA. I would have liked more information on these results :
a. Do women have change their mind between the 2 waves?
b. If yes, how many and in which context?
c. What is the potential impact on the results?
This information was dichotomised: what was the cut-off?
5. Line 251: Authors detailed their confounders: was GRA a confounders? In this case this is wrong because it is the characteristic evaluate in the study between the two groups so it should not be used as a confounder I their statistical analysis.
6. Line 255: tables 1, 2 and 3 should not be introduced in this methodology section.
Results:
1. For table 2 and 3, this way of result’ presentation is really not easy to read and understand for readers. The total number of respondents should be place at the top of the column. All model should be detailed one by one at the end of the table.
2. Figure 1 and 2 are easiest to understand quickly. The number of respondents in each category could be specified.
3. From line 324 to 334: this part should be replaced in the discussion.
Discussion:
1. From line 361 to 373: this part is the same idea that the paragraph 324-334. This 2 parts should be merged.
2. Line 392: there is a space.
3. There is no discussion about the repartition of domestic work that changes slowly. Line 403: the policy should not focus only on housewives with the presence of children but on whole family. The place of the partner/dad is never discuss whereas he is also an important part of this evolution.
Author Response
REVIEWER 1
Comment 1
“#Abstract:
It is well structured. Perhaps, the authors can detailed the number of participants in the study at this moment.
Response: We appreciate the reviewer’s suggestion. We added the number of participants in this study to abstract (line 23 to line 25).
Comment 2
“#Introduction:
- Line 36 : I think this is the “work conditions” of the whole working people (men or women) that have become more employee-friendly. I don’t think there are any labour rights reserved for women.
Response: Many thanks for the comments. Following your comments, we have revised this sentence (line 35-37)
Comment 3
“#Literature review:
This part should not exist in this type of epidemiology manuscript. These finding and empirical arguments must be replaced and shortened in the introduction or discussion if necessary.
- Line 165 : Authors propose 3 general questions :
-I think this is the work objectives which should be detailed by the main objective and the secondary ones.
-There is only 2 research questions detailed.
Response: We appreciate the reviewer’s suggestions. We replaced the part of the Literature Review with a part of the Theoretical Bases and moved some findings and arguments to the Introduction and Discussion. In addition, we excluded the general research questions section and summarised the assumptions by the end of the Theoretical Bases section (line 134 to line 139). Moreover, we added section 1.2 to summarise the key findings and contribution of the study(line 141 to line 160).
Comment 4
“#Methodology:
Authors explain us how the sample was made but we don’t know :
How people were contacted and how they are responded to?
How many total people were excluded from each analysis?
because 1,222 respondents above 40,000 people this is a very small part of the whole initial sample.
Response: We appreciate the reviewer’s suggestion. A flowchart has been made and shown on page 4, which presents the data-cleaning process step by step (please see lines 178 to line 179).
Comment 5
This is not usual to introduce a table of results in the methodology section. This table should be replace in the result section. Moreover this table gave us 2 information which should be placed in different results: the weighted sample and some descriptive ones. There is huge differences for two results: age groups and marital status. At this time of the manuscript, we don’t know how the two groups were made: traditional women and egalitarian women.
Response: We appreciate the reviewer’s suggestions. Table 1 of the descriptive analysis was moved to the Results section. Regarding the weighted sample descriptive statistics, we added the sample weight in the process of descriptive statistics to adjust the sampling bias, which is a very common operation in social statistics. Thus, the results of our sample descriptive statistics are nationally representative. Regarding the group setting, it was placed in the later section (line 269).
Comment 6
From line 204 to 216: this point is already explained line 185-191. This 2 two must be merged.
Response: We appreciate the reviewer’s suggestion. At the end of the Data and Sample section(originally from lines 185 to 191), we aimed to demonstrate the reason why we chose “paid employment & self-employ” rather than other job statuses in wave 4. In the section of Measures, for the description of Independent variables (originally from lines 204 to 216), we aimed to demonstrate the way that we generated this variable. We have already merged these two paragraphs appropriately(line 163 to line 177, and line 191 to line 201) .
Comment 7
-
Line 241: Authors explain us how they dichotomised GRA. I would have liked more information on these results :
- Do women have change their mind between the 2 waves?
- If yes, how many and in which context?
- What is the potential impact on the results?
This information was dichotomised: what was the cut-off?
Response: Many thanks for your comments. Firstly, yes, there are some cases that changed their gender role attitudes between the two waves. Specifically, around 84% of the cases remained in the same gender role attitudes groups across the waves. Secondly, we actually used the average of respondents’ gender attitudes across two waves rather than just one wave, which can partially avoid the potential bias caused by their changes in attitudes. Thirdly, drawing on many studies on gender role attitudes and mental health in the UK, we dichotomised the variable (GRA) by using the cut-off point 2.5/5 (please see the line 203 to line 220).
Comment 8
Line 251: Authors detailed their confounders: was GRA a confounders? In this case this is wrong because it is the characteristic evaluate in the study between the two groups so it should not be used as a confounder I their statistical analysis.
Response: We appreciate the reviewer’s suggestion. We corrected the name of GRA and the presence of children to the final moderators selected in this study (please see the line 203 to 220)
Comment 9
Line 255: tables 1, 2 and 3 should not be introduced in this methodology section.
Response: We appreciate the reviewer’s suggestion. We moved all these tables to the Result section.
Comment 10
Results:
- For tables 2 and 3, this way of results presentation is really not easy to read and understand for readers. The total number of respondents should be placed at the top of the column. All models should be detailed one by one at the end of the table.
Response: We appreciate the reviewer’s suggestions. We have formatted the tables by following the format in many papers published in IJERPH.
Comment 11
Figures 1 and 2 are easiest to understand quickly. The number of respondents in each category could be specified.
Response: We appreciate the reviewer’s suggestion.
Comment 12
From line 324 to 334: this part should be replaced in the discussion.
Response: We appreciate the reviewer’s suggestion. We merged this part to the discussion part (please see the line 325 to 355).
Comment 13
Discussion:
- From line 361 to 373: this part is the same idea that the paragraph 324-334. This 2 parts should be merged (please see the line 331 to 361).
Response: We appreciate the reviewer’s suggestion. We merged these two parts.
Comment 14
- Line 392: there is a space.
Response: We appreciate the reviewer’s suggestion. We deleted the space.
Comment 15
There is no discussion about the repartition of domestic work that changes slowly. Line 403: the policy should not focus only on housewives with the presence of children but on whole family. The place of the partner/dad is never discuss whereas he is also an important part of this evolution.
Response: We appreciate the reviewer’s suggestions. By the end of the discussion section, we added a brief discussion about the study’s contribution and implications (please see the line 372 to 375).
Reviewer 2 Report
Purhaps you could use this data along side a policy analysis to understand the evolution of mental health supports for women.
Is the target audience for this article policy makers?
The use of housewives and gender as a binary is dated. Also discussing women with children as only being in heterosexual relationships is limiting.
Discussing mental health has implications beyond policy, perhaps for mental health professionals, employee assistance programs, healthcare.
This data is a bit dated as well. What can we really learn from it that we don’t already know.
Author Response
REVIEWER 2
Comment 1
“Purhaps you could use this data along side a policy analysis to understand the evolution of mental health supports for women”.
Response: Many thanks for your comments. The authors of the study are not experts in policy analysis. We aim to contribute to public health policies by offering empirical evidence about the associations between housewives’ transition to paid employment and mental health with considering within-gender differences and the role of parenthood.
Comment 2
“Is the target audience for this article policymakers?”.
Response: Many thanks for this comment. Yes, the study’s findings contribute to the literature on occupational well-being and public health by highlighting the need for a more gender role attitudes-sensitive design in promoting women’s labour participation.
Comment 3
“The use of housewives and gender as a binary is dated. Also discussing women with children as only being in heterosexual relationships is limiting”.
Response: Many thanks. Your concern is insightful, but we occupy a traditional gender perspective due to the fact that traditional gender norms still constrain the labour market in the UK.
Comment 4
“Discussing mental health has implications beyond policy, perhaps for mental health professionals, employee assistance programs, healthcare.”
Response: We appreciate the reviewer’s comments. The study’s findings can contribute to occupational behaviours and healthcare literature.
Comment 5
“This data is a bit dated as well. What can we really learn from it that we don’t already know.”
Response: We appreciate the reviewer’s comments. Firstly, it is nationally representative data, which means the applicability of the study’s conclusion is much bigger than studies that adopted organisational-level data. Secondly, within-gender differences (gender role attitudes) are rather absent in current studies due to the limitation of many other datasets. Thirdly, the study’s findings about the intersection of within-gender differences and parenthood are novel.
Reviewer 3 Report
I found the paper generally interesting and pushes the conversation around the impact of women's gender roles on work and their mental health. However, authors need to address the below queries to further strengthen the paper.
1. Fix sentence on line 44 - from "with considering......"
2. Fix sentence from line 54 - from "women do not necessarily be satisfied......"
3. It might be useful to give a brief explanation of what gender role attitudes are and its relevance. Line 60
4. It might be useful to introduce what those within gender difference are and why it is important to extend the understanding to this level. Line 65
5. Better to capture it as "we integrated the literature on...." instead of using "we tend to integrate..." which may suggest that this is something you are regularly or frequently doing. Line 70
6. "On the one hand, there is a research tradition highlighting the benefits of employment to mental health" line 90. This claim needs to be referenced.
7. Literature review – from line 91 - While Jahoda, 1982 is a good reference point, a more recent reference would better strengthen the case for this direction. Find a recent paper to show that while this was flagged in 1982, it is still a case in recent times, hence the importance.
8. Jahoda (1982) needs proper in-text citation. Line 94.
9. Literature review – line 110: What are these specific traditional gender roles? Clarify what those are.
10. 3.2 Measures: "For the sake of clarity, this research follows the practice of previous research [42] and reverses the scale so that a higher score indicates greater mental health". Line 201.
So, does this mean that a higher score translates to poorer or better mental health? Greater mental health isn't well understood. Please make it clear so that your results can better interpreted.
11. Discussions and conclusion – from line 361. I can't help wondering if there was a distribution on the ages of the children as it could be argued that women with younger children (aged 0-6yrs) may have more demanding roles that those with older children. This is also very likely to play into the outcomes. Just something to think about.
12. Fix line issues from line 391 - 393.
13. It is unclear where the concluding paragraph starts. Introduce a new paragraph to conclude.

Author Response
REVIEWER 3
Comment 1
Fix sentence on line 44 - from “with considering......””.
Response: We appreciate the reviewer’s suggestions. We have fixed this sentence (please see the line 45 to 50).
Comment 2
Fix sentence from line 54 - from “women do not necessarily be satisfied......”
Response: We appreciate the reviewer’s suggestions. We have fixed this sentence (please see line 56 to 60).
Comment 3
It might be useful to give a brief explanation of what gender role attitudes are and its relevance. Line 60
Response: We appreciate the reviewer’s suggestions. We have explained the concept of gender role attitude further with references (please see the line 65 to 71).
Comment 4
It might be useful to introduce what those within gender difference are and why it is important to extend the understanding to this level. Line 65
Response: We appreciate the reviewer’s suggestions. This section has been revised with more details about the importance of understanding within-gender differences (please see the line 65 to 71).
Comment 5
Better to capture it as “we integrated the literature on....” instead of using “we tend to integrate...” which may suggest that this is something you are regularly or frequently doing. Line 70
Response: Many thanks for these suggestions. We have revised this sentence (please see line 76 to 78).
Comment 6
“On the one hand, there is a research tradition highlighting the benefits of employment to mental health” line 90. This claim needs to be referenced.
Response: We appreciate the reviewer’s suggestions. We have revised this sentence (see lines 80-82).
Comment 7
Literature review – from line 91 - While Jahoda, 1982 is a good reference point, a more recent reference would better strengthen the case for this direction. Find a recent paper to show that while this was flagged in 1982, it is still a case in recent times, hence the importance.
Response: We appreciate the reviewer’s suggestions. We added a sentence to bridge the theoretical predictions of Jahoda’s latent deprivation theory and current evidence on the benefits of employment (see lines 88 to 92). Most of the studies mentioned in this section tested and had conclusions that support the predictions of Jahoda’s latent deprivation theory.
Comment 8
Jahoda (1982) needs proper in-text citation. Line 94.
Response: We appreciate this comment. We have added proper in-text citation in this sentence (line 88 to 92)
Comment 9
Literature review – line 110: What are these specific traditional gender roles? Clarify what those are.
Response: We appreciate the reviewer’s suggestions. We added a further explanation about traditional gender roles in this section (lines 105 to 110).
Comment 10
3.2 Measures: “For the sake of clarity, this research follows the practice of previous research [42] and reverses the scale so that a higher score indicates greater mental health”. Line 201.
So, does this mean that a higher score translates to poorer or better mental health? Greater mental health isn’t well understood. Please make it clear so that your results can be better interpreted.
Response: We appreciate the reviewer’s suggestions. We clarified that in this study, a higher score of GHQ indicates better mental health status (see line 187 to 188).
Comment 11
Discussions and conclusion – from line 361. I can’t help wondering if there was a distribution on the ages of the children as it could be argued that women with younger children (aged 0-6yrs) may have more demanding roles that those with older children. This is also very likely to play into the outcomes. Just something to think about.
Response: Many thanks for these insightful comments! Yes, we totally agree with your ideas that women with younger children might suffer more demanding roles and this can lead to potential group differences in the associations. We will be very happy to investigate these differences in our future studies.
Comment 12
Fix line issues from line 391 - 393.
Response: We appreciate the reviewer’s suggestions. We have fixed these line issues (please see line 388 to line 392).
Comment 13
It is unclear where the concluding paragraph starts. Introduce a new paragraph to conclude.
Response: We appreciate the reviewer’s suggestions. We have introduced a new paragraph to conclude (start from line 377).
Round 2
Reviewer 2 Report
This paper is fine for publication based on the outlined evaluation criteria. Overall relevance to the current context is low because of the age of the data and lack of novel ideas. I recommend engaging with critical perspectives and comparison when using older data sets in the social sciences.
Reviewer 3 Report
I am satisfied with the amendments made.